# Phytotoxicity of Essential Oils: Opportunities and Constraints for the Development of Biopesticides. A Review

**DOI:** 10.3390/foods9091291

**Published:** 2020-09-14

**Authors:** Pierre-Yves Werrie, Bastien Durenne, Pierre Delaplace, Marie-Laure Fauconnier

**Affiliations:** 1Laboratory of Chemistry of Natural Molecules, Gembloux Agro-Bio Tech, University of Liège, 5030 Gembloux, Belgium; marie-laure.fauconnier@uliege.be; 2Soil, Water and Integrated Production Unit, Walloon Agricultural Research Centre, 5030 Gembloux, Belgium; b.durenne@cra.wallonie.be; 3Plant Sciences, Gembloux Agro-Bio Tech, University of Liège, 5030 Gembloux, Belgium; pierre.delaplace@uliege.be

**Keywords:** essential oils, phytotoxicity, mode of action, biopesticides

## Abstract

The extensive use of chemical pesticides leads to risks for both the environment and human health due to the toxicity and poor biodegradability that they may present. Farmers therefore need alternative agricultural practices including the use of natural molecules to achieve more sustainable production methods to meet consumer and societal expectations. Numerous studies have reported the potential of essential oils as biopesticides for integrated weed or pest management. However, their phytotoxic properties have long been a major drawback for their potential applicability (apart from herbicidal application). Therefore, deciphering the mode of action of essential oils exogenously applied in regards to their potential phytotoxicity will help in the development of biopesticides for sustainable agriculture. Nowadays, plant physiologists are attempting to understand the mechanisms underlying their phytotoxicity at both cellular and molecular levels using transcriptomic and metabolomic tools. This review systematically discusses the functional and cellular impacts of essential oils applied in the agronomic context. Putative molecular targets and resulting physiological disturbances are described. New opportunities regarding the development of biopesticides are discussed including biostimulation and defense elicitation or priming properties of essential oils.

## 1. Introduction

Essential oils (EOs) have been used historically in the food and perfume industries and are extracted from various plant organs (flowers, leaves, barks, wood, roots, rhizomes, fruits and seeds) through steam distillation, hydro-distillation and cold expression for citrus. These natural products are mainly composed of volatile organic compounds (VOCs), having a high vapor pressure at room temperature and belonging mainly to the phenylpropanoid and terpenoid families. Briefly, terpenes are classified according to the number of isoprene sub-units: two for monoterpene (C_10_H_16_) and three for sesquiterpene (C_15_H_24_). Oxygenated terpenes or terpenoids also contain additional functional groups such as alcohol, carboxylic acid, ester, etc. [1], and phenylpropanoids are produced from L-phenylalanine through deamination by phenylalanine ammonia-lyase [2].

Many research studies have been undertaken on the use of EOs in more sustainable agronomic practices. In this regard, numerous findings have described the strong biopesticidal potential of EOs thanks to their antibacterial [3], antifungal [4], insecticidal [5], acaricidal [6], nematicidal [7] and herbicidal activities [8]. Included under the Generally Recognized as Safe (GRAS) product categories of the United States Food and Drug Administration, the impact of EOs on human health and ecosystems seems to be lower compared to synthetic plant protection products (PPP). Biocidal actions of EOs can be specific, and therefore their use could be compatible with integrated pest management (IPM) [9].

The application of EOs is, however, subject to a major constraint. They may present phytotoxic properties to untargeted plants such as crops. The most effective EOs in pest control are phytotoxic too, and considerable precautions are required regarding product formulation (unless the objective is the formulation of a total herbicide) [10]. Empirical tests for commercial EOs are commonly realized on major crops [11]. However these strategies have led to poor knowledge relating to other biological systems [12]. Many parameters determine this impact, such as the application mode (root watering, aerial spraying or injection in the vascular system), the plant organs targeted, the phenological stage (seed, plantlet or mature plant), the physiological state and product formulation. As illustrated by the opposing claims regarding the presence or absence of phytotoxicity of *Mentha pulegium* (pennyroyal) EOs towards *Cucumis sativus* (cucumber) and *Solanum lycopersicum* (tomato), it is necessary to gain insight into the molecular mechanism involved in order to design suitable biopesticides [13,14,15].

Phytotoxicity can be defined as a negative impact on plant growth or plant fitness and can be linked to cellular dysfunctions. Physiological impairment can be observed through integrative measurements of stress, for example on the photosynthetic apparatus. However, determination of the primary site of action is much more challenging. Diverse phytochemical products have been demonstrated to influence several physiological processes of growth and development in plant cell division and root elongation [16]. Blends of natural plant compounds often have numerous mechanisms of action, making them very efficient at acting on a plant’s primary metabolism. It therefore seems most important to gain an insight into the physiological impact of EOs on plant crops to design proper bioassays and efficient biopesticides. Avoiding residual phytotoxicity, which is currently an underestimated constraint in the field, will allow the broader application of EOs [17]. However even if some processes seem to be inhibited in a dose-dependent manner, a concentration below the phytotoxic threshold could also stimulate the plant, a phenomenon referred to as biostimulation. New opportunities arising from this biostimulation and elicitation of defenses will be discussed in this review.

All the mechanisms involved in the phytotoxicity of EOs cannot be easily interpreted individually [18]. This review aims to discuss the latest putative molecular targets (mode of action) involved in plant metabolism with a physiological approach including water status alteration, membrane interaction/disruption, reactive oxygen/nitrogen species induction, genotoxicity and microtubule disruption, mitochondrial respiration or photosynthesis inhibition and enzymatic or phytohormones regulation. The different mechanisms presented throughout this review have been graphically summarized in Figure 1.

## 2. Essential Oils’ Cellular and Physiological Impacts

### 2.1. Essential Oils’ Translocation

Essential oil constituents (EOC) must access specific targets in order to carry out the physiological impact previously listed within a plant. Numerous publications describe the VOCs released by plants [19,20,21]. However little is known about their cellular entrance and translocation in plant organisms in the case of a systemic effect.

When sprayed, the first interaction occurs with the cuticular wax components of the leaves. In fact, the cuticle is considered to be the plant’s first barrier to molecule penetration. The interaction between monoterpene with epicuticular waxes and stomata will be further described. Briefly, once it has entered through the stomata opening by gas exchange or diffusion through the waxy cuticle, each EOC is partitioned into the gas phase and liquid phase following a defined ratio determined by Henry’s law. The liquid phase is materialized by the cell wall in which EOC accumulates. Compounds then diffuse to the cytosol following their oil/water partition coefficients [22]. Finally, active transport should also be considered as has been demonstrated for emissions [23].

Regarding root uptake, a study with radio-labelled thymol demonstrates the translocation of monoterpenes in citrus trees. However, the determination of the mechanism was beyond the scope of the study, although the authors suggest it could be similar to that for ethylenediaminetetraacetic acid (EDTA) [24].

### 2.2. Water Status Alteration

Depending on the mode of application (aerial or root), two different phenomena have been suggested for disturbing the water status of plants after treatment with EOs.

The deleterious effect of monoterpene (camphor and menthol) on cuticular wax and stomatal closure inhibition has been observed [25]. These two effects act synergistically on plant transpiration leading to guard cell disruption and desiccation. Interestingly, an opposite growth promoting effect is described for *Arabidopsis thaliana* during short vapor exposure to these terpenes. The molecular mechanism responsible for this prevention of stomatal closure is mediated through modification in the cytoskeleton and especially in the actin filament. Furthermore, stress symptoms appear together with a change in gene expression [26]. The amount of leaf epicuticular waxes determines the sensitivity of crop seedlings and weed species [27].

Water status alteration of plants was also observed after root watering application with citral, a mixture of two monoterpene isomers neral and geranial [28]. In a similar study with the sesquiterpene trans-caryophyllene, the authors suggest that this alteration could be responsible for the oxidative burst and a strong proline accumulation due to its osmo-regulative function [29].

### 2.3. Membrane Properties and Interactions

After entering the intercellular space through the mesh of the cell wall, EOCs directly solubilize within the plasma membrane depending on their physical properties, particularly their vapor pressure and molecular mass. Their specific accumulation was demonstrated to modify the lipid packing density, membrane-bound enzymes and ion flux [30].

This interaction can lead to a reversible depolarization of the membrane potential (Vm) and to membrane disruption [31]. Furthermore, stronger membrane depolarization occurs for more water soluble monoterpenes presenting a low octanol/water partition coefficient (Kow). A change in the polarization state implies ion mobility through the membrane. A drastic entrance of Ca^2+^ in the cytosol is triggered by opening the calcium channel. Ca^2+^ is known to be largely involved in cellular signaling. It performs allosteric regulation of many enzymes and proteins. Moreover, Ca^2+^ is an intracellular second messenger of signal transduction pathways and gene expression. Finally, the increase in Ca^2+^ concentration can lead to an oxidative burst [32].

Studies on artificial monolayer membranes of dipalmitoyl-phosphatildylcholine describe the penetration of monoterpenes such as camphor, cineole, thymol, menthol and geraniol, which affect the vesicles topology [33]. Similar work on model bilayer interactions with related monoterpenes, including limonene, perillyl alcohol and aldehyde, demonstrates the diffusion across the membrane and an ordering effect on the lipid bilayer [34]. More recently, novel molecular techniques of dynamic interaction were applied to study the interaction between citronellal (monoterpene), citronellol (monoterpene) and cinnamaldehyde (phenylpropanoids) with a biomimetic membrane [35]. Briefly, the in silico insertion model predicted different behaviors between the two classes (monoterpenes and phenylpropanoids). These predictions were confirmed using in vitro biophysical assays. Citronellal and citronellol interaction with the model membranes was demonstrated without permeabilizing it, while cinnamaldehyde did not interact with the model membrane. This suggests two different mechanisms of action: (i) the modification of lipid bilayer organization by monoterpenes and (ii) the interaction with membrane receptors for phenylpropanoid pathway metabolites.

Associated with the modification of membrane properties, a change in the membrane’s composition also occurs. In fact, an increase in unsaturated fatty acids was demonstrated following application of monoterpenes such as 1,8-cineole, geraniol, thymol, menthol and camphor [36]. Quantitative and qualitative changes in most abundant free and esterified sterols (sitosterol, stigmasterol, and campesterol) and phospholipid fatty acids (16:0, 16:1, 18:0, 18:1, 18:2, 18:3) were also highlighted in a study investigating the effect of the same monoterpenes [37]. This results in an increase in the percentage of unsaturated fatty acid (PLFAs) and stigmasterol. Interestingly, alcoholic monoterpenes seem to have a different mode of action affecting more unsaturated fatty acid and stigmasterol leading to seedling growth interferences.

### 2.4. Reactive Oxygen and Nitrogen Species Induction

Reactive oxygen species (ROS) are essential in cellular signaling. They can be produced in various locations in plant cells such as in the chloroplast, the peroxisome, the mitochondria and in the endoplasmic reticulum. ROS are very reactive compounds that in excess lead to the degradation of macromolecules such as lipids, carbohydrates, proteins and DNA [38].

Oxidative burst or generation of ROS has long been proposed as one of the main mechanisms of action of phytotoxins [39]. We know that the uncoupling of photosynthesis and respiration leads to the production of superoxide radicals (O^2−^), which are transformed into hydrogen peroxide (H_2_O_2_) by the superoxide dismutase. Moreover, the reaction with transition metal triggers a reduction of H_2_O_2_ to OH^.^, another very reactive species [40].

Oxidative stress was acknowledged after treatment with α-pinene through hydrogen peroxide, proline and the lipid peroxidation product malondialdehyde (MDA). Moreover, an antioxidant enzyme activity assay (superoxide dismutase, catalase, ascorbate, peroxidase, guaiacol peroxidase and glutathione reductase) was also performed in the roots. The oxidative stress generated by these ROS leads to membrane lipid peroxidation and ultimately to membrane disruption launching the programmed cell death. These membrane disruptions are evidenced via electrolyte leakage (EL) and vital staining [41].

In a similar experiment determining germination and growth inhibition by β-pinene EL, lipid peroxidation and lipoxygenase (LOX) activity were assessed. The result showed a strong increase in EL, dienes and H_2_O_2_ content and the authors suggest that despite an increase in the activity of ROS scavenging enzymes, root membrane integrity was lost [42]. Later on, they studied the early ROS generation and activity of the antioxidant defense system in the root and shoot of hydroponic wheat. The damaged was more severe in the root and a higher lipoxygenase activity was observed in parallel with accumulation of MDA [43]. The up-regulation of LOX activity has been observed for citronellol as well and the authors suggest that its hydroperoxide derivatives may destroy the membrane [44].

EOs inhibiting the growth of tested plants via ROS overproduction leading to oxidative stress and degradation of membrane integrity was evidenced via increased levels of MDA and EL, and decreased levels of conjugated dienes were demonstrated for other EOs such as *Pogostemon benghalensis* [45], *Monarda didyma* [46] and *Artemisia scoparia* [47].

Secondary effects of ROS generation include depigmentation of cotyledons in *A. thaliana* by *Heterothalamus psiadioides* EOs. The effects are here observed in a dose-dependent manner and in very small amounts. The authors also suggest that alteration on auxin levels occur as a secondary effect. Exogenous addition of antioxidants did not reverse effects on adventitious rooting, indicating that damages were too severe [48].

The generation of ROS, one of the most prevalent plant responses to stress, is described in direct response to the application of EOs. However, it is unlikely to be the main mechanism of toxicity but rather an indirect consequence resulting from LOX activity, chloroplast or mitochondria alteration [38]. The fundamental involvement of ROS in stress signaling as well as their interaction with other signaling components such as transcription factors, plant hormones, calcium, membrane, G-protein and mitogen-activated protein kinases need to be highlighted [49]. These interactions may explain many of the numerous physiological impacts induced by EOs’ application in plants. Moreover, after treatment with α-farnesene, they also observed the induction of nitric oxide production, a reactive nitrogen species (RNS) associated with an oxidative burst [38].

### 2.5. Photosynthesis Inhibition

Photosynthesis inhibition has also been proposed as one of the putative modes of action of EOs. While the impact of certain allelochemicals on photosynthesis is well established, for instance quinone, this is not the case for EOs where numerous mechanisms have been proposed. Direct ROS-mediated disruption through oxidation of photosystem II (PSII) protein has been suggested to inhibit photosynthesis as suggested by the increase in the proline content, whose function is to accept electrons to protect the photosystem [50]. The effect of β-pinene on the chloroplast membrane has long been demonstrated by the inhibition of the electron transport of PSII [51,52].

Numerous studies report a decrease in the photosynthetic pigments namely chlorophylls (a and b) and carotenoids after treatments with EOs in a dose-dependent way [53,54,55]. This can result from a direct pigment photo-degradation or from a decrease in *de novo* synthesis. Plants have developed a non-photochemical quenching (fluorescence) strategy to avoid the ROS production resulting from this photo-inhibition. The decrease in carotenoid content could explain a higher fluorescence emission and a decrease of the PSII performance due to some damage to the complex antenna via ROS production and lipid peroxidation [56].

*Artemisia fragrans* EO impacts on the photosynthetic apparatus of perennial weed *Convolvulus arvensis* were studied using the most important chlorophyll fluorescence parameters. Increase in minimal fluorescence level (F0) implies a restriction in the PSII transport chain. The decrease in maximum quantum yield of PSII (Fv/Fm) results from photosystem inactivation (photo-damage) and/or a blockade in electron transport. PSII electron transport chain state (φPSII) reduction in plants treated with EOs restricts the non-cyclic electron transport chain. The last two parameters represent energy used in photochemical quenching (qP) and non-photochemical quenching (NPQ). qP decreases following concentration of EOs whereas NQP increases. Taken altogether, these results imply that the excited energy was not used in photosynthesis due to photosystem degradation by EO treatment [57].

Two specific fluorescence parameters QYmax (a maximum quantum yield of PSII photochemistry) and Rfd (a fluorescence decrease ratio) have even been proposed as early predictors of broccoli plant response treatment to clove oil [58].

Moreover, in a study of photo respiratory pathway alteration by *Origanum vulgare* EOs in *A. thaliana*, Araniti et al. [59] suggested that alteration of glutamate and aspartate metabolism leads to leaf chlorosis and necrosis. Glutamine synthetase is crucial to incorporate ammonia in organic compounds and may be a molecular target of *O. vulgare* EO. Finally, ammonia accretion has direct inhibiting properties on PSI and PSII due to its bonding with the oxygen-evolving complex. In addition, the decrease in pH gradients across membranes is able to uncouple photophosphorylation.

### 2.6. Mitochondrial Respiration Inhibition

Mitochondrial respiration inhibition is another putative target in the cellular mode of action of EOs. Monoterpene treatment has long been reported to decrease respiratory oxygen consumption in whole plants, dissected organs and isolated mitochondria for 1,8-cineole [60] and juglone [61].

The effect of monoterpenes has been well documented on isolated mitochondria, on germination and on primary root growth of maize [62]. Briefly, the authors demonstrated that α-pinene triggers two different mechanisms which are the uncoupling of oxidative phosphorylation and the inhibition of electron transfer. This action drastically decreases adenosine triphosphate (ATP) production and the authors suggest it occurs following unspecific disruption in the inner mitochondrial membrane [63,64]. The mode of action of other monoterpenes such as camphor and limonene have been investigated. They respectively cause mitochondrial uncoupling and act on ATP synthase or on adenine nucleotide translocase complexes [63,65].

Accessibility to mitochondria in vivo can strongly affect phytotoxicity. A study performed using soy hypocotyl showed that the effect on mitochondria alone did not fully explain the resulting phytotoxic effect. Absence of correlation between respiratory inhibition in mitochondria and seed germination or root growth treated with α-pinene and limonene suggest that their inhibition properties are probably dependent on their ability to permeate intracellular compartments [65].

Furthermore, the description of the cytochrome-oxidase pathway inhibition highlights the fact that this inhibition is likely to increase mitochondrial reactive oxygen species and membrane lipoperoxidation as demonstrated by increased concentrations of lipoperoxide products, activation of lipoxygenase and antioxidant enzymes [66].

Microscopic evaluation highlights the drastic reduction in the number of intact organelles among which mitochondria and membranes disrupt nuclei, mitochondria and dictyosomes [67]. This mitochondrial membrane deleterious effect leads to a decrease in energy production and ROS generation affecting numerous biochemical processes and cellular activities as observed for tobacco BY-2 cells treated with 1,8-cineole [68,69].

### 2.7. Microtubule Disruption and Genotoxicity

Vapor exposure of citral at µmolar concentrations completely depolymerizes microtubules without any damage to the plasma membrane [70]. Results suggest an in vitro dose/time relationship for microtubule disruption whereas the actin filament remained intact. Finally mitotic microtubules were more damaged than the cortical ones, leading to impairment in the mitosis process [71].

To determine whether the microtubule impact results from direct depolymerization or from indirect phytohormones balance modification, Graña et al. [72] studied the short- and long-term effects of citral application in the plant model *A. thaliana*. Auxins (indole 3-acetic acid) polar transport is rapidly inhibited and ethylene content increases. These two hormones have numerous points of interaction and are essential for microtubule organization, which leads to a long-term disorganization of cell ultra-structure. Citral-treated samples present a large number of Golgi complexes together with a thickening of the cell wall. Those phenomena affect cell division and intracellular communication in the long term.

More recently, Chaimovitsh et al. [73] studied microtubule and membrane damages for a large number of terpenes and further demonstrated the difference in their mechanisms of action. In fact, they observed strong microtubule depolarization for limonene and (+)-citronellal and moderate microtubule depolarization for citral, geraniol, (−)-menthone, (+)-carvone and (−)-citronellal. Moreover, many compounds lacked antitubular activity such as pulegone, (−)-carvone, carvacrol, nerol, geranic acid, (+)/(−)-citronellol and citronellic acid. Furthermore, they demonstrated enantioselectivity of microtubule disruption for citronellal and carvone, the (+) enantiomers being more effective. They compared this antitubular activity with the membrane disrupting properties and found that citral did not cause membrane disruption. Carvacrol induced membrane leakage, and limonene both depolymerized microtubules and induced membrane leakage. Finally, through in vivo quantification of applied monoterpene they discover the biotransformation of citral (i) and limonene (ii) to (i) nerol and geraniol and (ii) carvacrol, respectively. This conversion explains the dual mode of action of limonene in both the membrane and microtubule. Dual mode of action was recently highlighted for menthone in tobacco BY-2 plant cells and seedlings of *A. thaliana* [74].

Concerning direct genotoxicity, numerous chromosome abnormalities have been observed, such as sticky chromosome, chromosome bridges, spindle disturbance, c-mitosis and bi-nucleated cells in root tip cells after treatment with EOs of *Schinus terebinthifolius*, *Citrus aurantiifolia*, *Lectranthus amboinicus*, *Mentha longifolia* and *Nepeta nuda*. The damaging reaction of EOs on the chromatin organization could lead to chromosome bridges or sickness and ultimately to apoptosis. Interestingly, different results for EOs with the same principal terpene suggest that there is a synergic interaction between major and minor compounds [75,76,77,78,79].

Another mito-depressive activity of EOs could be mediated by the inhibition of DNA synthesis. It was effectively demonstrated by Nishida et al. [80] that monoterpenes are able to hinder organelle and nuclear DNA synthesis. Direct damage to DNA has been highlighted through the effect of EOs on head and tail DNA. Although the mechanisms behind this are still vague, authors suggest that ROS following EO treatments may be responsible for the genotoxic effect [81].

### 2.8. Enzymatic Inhibition and Regulation

Beside glutamine synthetase as a particular enzymatic target of EOs, studies suggest direct or indirect inhibition of specific enzymes as a putative mode of action. For example, a first case is related to the long known potato tuber bud dormancy inhibition using peppermint oil. A decrease in the activity of 3-hydroxy-3-methylglutaryl Coenzyme A reductase (HMGR; E.C. 1.1.1.34), a key-enzyme in the mevalonate pathway, was observed but without explanation at the transcriptional level [82,83].

Rentzsch et al. [84] demonstrated a specific monoterpene interaction with gibberellin (GAs) signaling at the dose-, tissue- and gene-level during dormancy release and sprout growth. They also described a typical case of biostimulation. At low concentrations, peppermint essential oil and carvone promote bud sprouting and dormancy release, whereas at high concentrations they completely inhibit it. They demonstrated that dormancy release is associated with tissue-specific α- and β-amylase modulation and that EOs could affect this modulation. Indeed, at low concentration, amylase expressions were modulated by carvone through specific enhancement of a-AMY2 gene transcription by interacting with its transcription factor. This was not the case for peppermint EOs, for which they proposed interaction with specific components of the GAs signaling pathway that enhanced the GAs-mediated responses [84].

These enzyme modulating activities have been reported for other compounds such as β-pinene reduction of hydrolyzing enzyme (protease, α- and β-amylase) in rice seedlings. At the same time, peroxidases and polyphenol oxidase activity increases, suggesting their role in resistance against β-pinene-induced oxidative stress [53].

Strict inhibition phenomena have been proposed for cinmethylin, which is a synthetic analogue of 1,4 and 1,8-cineole through asparagine synthetase inhibition. Authors have suggested that benzyl ether moiety cleaved to generate toxophore that inhibits the enzyme. However due to an inability to reproduce these results in vivo afterwards, the authors decided to retract the paper. This illustrates well the difficulties in rigorously establishing a single molecular target [85].

Later another target was proposed for the herbicide cinmethylin, the tyrosine aminotransferase (TAT; EC 2.6.1.5). Indeed, TAT provides quinones for the prenylquinones pathway in the inner chloroplast membrane. Furthermore, plastoquinone is a cofactor in the carotenoid pathway. Therefore, the decrease in carotenoid resulting from this inhibition may trigger photo-oxidative degradation of chlorophyll and photosynthetic membranes, disturbing chloroplast function [86].

More recently, Abdelgaleil, Gouda and Saad [87] postulated that phytotoxicity of EOs could be mediated through carbonic anhydrase inhibition. Indeed, this enzyme plays a key role in the (de)carboxylation reaction involved in both respiration and photosynthesis and contributes to the movement of inorganic carbon to photosynthetic cells. Thus, CO_2_ content in these cells would decrease, leading to the formation of ROS by diverting a photosynthetic electron from CO_2_ [87].

### 2.9. Phytohormones and Priming of Plant Defence

A first evidence of the interaction with phytohormones has already been developed previously concerning the gibberellin (GAs). Two other interconnected hormones have been suggested as main targets, auxins and ethylene. Indeed, citral impacts the polar auxins transport, resulting in an alteration of its content, cell division and ultrastructure of *A. thaliana* root meristem seedlings cell [72]. Concentration balance between auxin and ethylene is responsible for root growth, radicle elongation and root hair formation. Citral was suggested as a promising herbicide with strong short term and long lasting toxicity. Similar results on polar auxin transportation were obtained with farnesene [88], which affects specific PIN-FORMED (PIN) protein. Furthermore, modification in PIN gene expression leads to a decrease in meristem size and a left-handed phenotype. Interestingly, a previous study reported an increase in the auxin content [56]. This loss of gravitropism was suggested to result from an alteration in the hormonal balance and stimulation of oxidative stress via ROS and RNS production interfering with cell division and cytokinesis through microtubule disruption altering root morphology.

Phytohormone balance is also involved in priming and plant defense induction mechanisms. Monoterpenoids are able to activate defense genes by signaling processes and Ca^2+^ influx causes by membrane depolarization, protein phosphorylation/dephosphorylation and the action of ROS [89]. This gene expression can either lead to priming (an accelerated gene-response to biotic stress) or direct defense elicitations.

Priming of plant defenses has already been acknowledged in agricultural practices, as for example exposure to mint volatiles, which enhanced transcripts levels of defense genes in soy through histone acetylation within the promoter regions [90]. This priming was stronger at mid-distance, implying a nonlinear relationship to concentration. Recently, priming against bacteria was observed in apple using thyme oil. Indeed, the authors noted a much stronger expression of pathogenesis-related (PR) genes PR-8 following *Botrytis cinerea* application [91].

Regarding elicitation of plant defense, resistance can either be constitutive with the systemic acquired resistance (SAR) or induced with the induced systemic resistance (ISR). There is large cross-talk between the two systems which rely on salicylic acid (SA) and jasmonate (JA) hormones.

Transcriptomic study following exposure to volatile monoterpenes myrcene and ocimene demonstrated that plants develop a similar response to that induced by methyl jasmonate (MeJA) [92]. Microarray profiling revealed the induction of several hundreds of transcripts annotated as stress or defense genes or transcription factor. Multiple stages of the octadecanoid pathway were present, and metabolite analysis demonstrates an increased level of MeJA in *A. thaliana* tissues.

The induction of SAR has also been acknowledged when using *Gaultheria procumbens* essential oil, which is composed almost only of methyl salicylate. To demonstrate the effectiveness of the EO, they inoculated GFP-labelled fungal pathogens and showed a strong reduction in its development, similar to commercial solution [93]. Thyme EO also triggers constitutive defense in tomato against grey mold and fusarium as demonstrated by phenolic compounds and peroxidase activity measurements. Furthermore, root application is more effective than foliar. The authors also suggest that an increase in peroxidase activity resulting from oxidative burst (ROS) is a precursor of phenolic compound accumulation. It seems that activation of a plant defense gene and secondary metabolite production can be attributed to Peroxidase-Mediated Reactive Oxygen Species production [94]. Moreover, induction of defense enzymes associated with SAR such as β-l,3-glucanase, chitinase and peroxidase activity, have been observed for different essential oil/constituents namely *Cinnamomum zeylanicum* oil/trans-cinnamaldehyde [95], Indian clove EO/eugenol [96] and citronella EO/citronellal [97].

## 3. Mechanism of Detoxification

Plants have evolved pathways to decrease the toxicity of allelochemicals released from neighbors and xenobiotics. These mechanisms can be summarized as the metabolization of phytotoxins or conjugation/sequestration followed by compartmentalization or emissions.

Reduction and esterification of aldehydes to their alcohols have been demonstrated for green leaf volatiles such (GLV) as (*Z*)-3-hexenal [98], but also as previously mentioned for monoterpenes such as citral to nerol and geraniol and limonene to carvacrol [73]. Similar reaction pathways were mentioned for citronellal by *Solanum aviculare* suspension cultures to menthane-3,8-diol, citronellol and isopulegol [99]. Wheat seeds exposed to EOs were also able to oxidize and reduce different terpenes, namely neral, geranial, citronellal, pulegone and carvacrol, to the corresponding alcohol and acids using non-specific enzyme systems. The authors have suggested that the reduction activity was catalyzed by non-specific dehydrogenase and oxidation by P-450-type enzymes [100]. Interestingly, part of the applied compound is degraded, as demonstrated by the impossibility to account for all the compounds supplied to the germinated seeds. Moreover, derivates are less toxic compared to parent compounds [100]. *Anethum graveolens* hairy root cultures biotransform two oxygen-containing monoterpene substrates, menthol or geraniol in 48 h to menthyl acetate, linalool, α-terpineol, citronellol, neral, geranial, citronellyl, neryl, geranyl acetates and nerol oxides [101].

Other detoxifying mechanisms rely on conjugation with carbohydrates, or glycosylation, to sequestrate VOC. Compared to the free aglycones, they present a higher solubility in water and a smaller reactivity, which facilitates their storage in the vacuoles and protects from aglycones toxicity [102]. Numerous studies demonstrate this glycosylation by *Eucalyptus perriniana* culture cell which converts thymol, carvacrol and eugenol into the corresponding β-glucosides and β-gentiobiosides [103]. Biotransformation products were isolated following administration of 1,8-cineole as well. Following the administration of camphor, seven new mono-glucoside products were isolated. Interestingly, the oxygen function was introduced before the glycosylation and ketone group reduction was observed [104]. (−)-fenchone administration delivered six new biotransformation products with specific regio- and stereoselectivity for the hydroxylation reaction [105]. Similar results were obtained for sesamol [106] and vanillin [107] as well.

Cell suspension of *Achillea millefolium* administrated with geraniol, borneol, menthol, thymol and farnesol converts these into several products and glycosylate, both the substrates and the biotransformation products. The decrease in glycosylated compounds afterwards implies that this glycolization mechanism is both used for detoxification and to convert VOC in readily usable forms to incorporate them in the metabolism [108].

This mechanism was also acknowledged *in planta* as demonstrated for (*Z*)-3-hexenol produced by plants under insect attack [109]. This glycolized form acts as a defense molecule against herbivores, and is accumulated for the sake of prevention of the next attack. A large number of plant families use glycolization as a common pathway of exogenous VOC plant perception. Similar results are observed for other types of alcohols including aromatic, aliphatic and terpene compounds [110].

Another sequestrating reaction consisted in the glutathionylation of GLV, which has been demonstrated for methacrolein whose gluthation conjugates have been isolated from vapor-exposed tomato [111]. α, β-unsaturated aldehydes also react with gluthation [112]. Overall, various processes have been developed by plants to detoxify and they are summarized in Figure 2.

## 4. Discussion and Conclusions

EOs physiological impacts have been and can be studied at the metabolomic [113], proteomic [114] and transcriptomic [115] levels and large amounts of untargeted data will emerge by grouping these techniques of research together. As phytotoxicity is either a goal (herbicide) or a constraint (other biopesticidal application or biostimulation), both parts will be discussed separately.

Regarding herbicidal application, cellular metabolism reactions are clearly involved in the phytotoxic properties of essential oils. The scientific community is making progress in identifying the cellular functions affected, such as photosynthesis, respiration, etc., and research is advancing in molecular target identification. Nevertheless, due to the many interconnecting pathways that are involved simultaneously, no clear distinction has appeared between the diverse chemical classes of EOs compounds. Most of them are grouped within one EO, which makes the unravelling of the specific mode of action a complex process. However, their effects can be distinguished between a general stress type response (ROS or osmotic related) compared to a more specific target (microtubule for example) leading to cellular impairment at a much lower concentration.

To demonstrate persistence and efficiency in the targeted biological system, medium- and long-term effects are most important. To answer these questions, it seems most interesting to deepen the study on the dynamics of the compounds and their fate in plant metabolism in regards to the capacity of the plant to metabolize, detoxify, sequestrate and compartmentalize. Phytotoxicity towards weeds without affecting the crop is essential to develop selective bio-herbicides. In this regard, the identification of other molecular mechanisms such as sugar and amino acid accumulation to prevent EOs stress seems promising as demonstrated in maize [113].

The last point relates to the composition of the EOs. High complexity of EOC needs to be characterized properly as hundreds of compounds sometimes occur [116]. Moreover, variability within the same genus or plant has been frequently observed depending on many parameters such as chemotype, climate, soil, exposure from one year to the next [117,118], sometimes leading to fundamentally different compositions [119]. However, even if fundamental interaction cannot be studied properly for hundreds of compounds, their diverse mechanisms of action can constitute a strong opportunity for synergistic effects and prevent adaptation by weed species. Interaction between different EOC can allow a reduction in the application, while still effectively preventing germination and weed growth [120].

On the other hand, the phytotoxicity of essential oil has long been considered as its main constraint regarding the development of other biopesticides (insecticides, fungicides, etc.) Phytotoxic consideration is currently often limited to the trade-offs of efficiency against the targeted pest versus visual innocuousness to the protected crop. As illustrated in Table 1, large variation occurs regarding the phytotoxic properties of EOs or their constituents depending on the application systems and mode of action considered.

Bioassays should ideally provide a range of toxic concentrations according to the mechanism involved in the toxicity process. Standardized methodologies/protocols to define the toxicity level of individual compounds as well as their blends are needed at the macroscopic or remote level and on a specific scale to allow prediction. It is always a question of targeting an applied plant model and then defining the toxicity levels in those specific application conditions. In this regard, in vivo redox and osmotic status sensor should be used as a specific marker of toxicity levels.

Other opportunities seem to arise at low concentrations far below the toxicity threshold, such as biostimulation [121] and priming or elicitation of defense mechanisms [91]. This elicitation of the systemic defense mechanism can also result in broader abiotic pest protection and be a pertinent agronomical strategy. However, limitations arise in regard to the allocation of resources (growth-defense trade-off) and reduced efficiency compared to a synthetic product. The same essential oils/constituents are sometimes mentioned to be phytotoxic at high concentrations and beneficial at low ones following a dose response concept. It has been proposed that these low doses simulate mild stress [122]. However, such threshold models as hormesis are still debated in biology and very little is known about the underlying mechanisms [123].

An additional consideration concerns the kinetic release of EOs. Indeed, their persistence and application methods are limited due to their low molecular weight, hydrophobicity and high volatility. To overcome these limitations, much work has been done regarding formulation techniques to allow a control release profile. A recent promising domain is the formulation of nano-emulsion using bio-based surfactants [124] as well as other encapsulation techniques [125].

A final constraint is the market approval by the different regulatory agencies throughout the world as well as economic considerations. Even if procedures are sometimes available for plant-based products such as GRAS, list 25b of the EPA [12] or the European Pesticide Regulation (EC) No. 1107/2009 [126], few active substances have been registered so far. Easier registration also leads to misevaluation regarding efficacy and safety for consumers. Indeed, in high concentrations, their use may be economically disadvantageous and exhibit undesirable phytotoxicity [127]. In fact, the mammalian toxicity (LD50) is >1000 mg kg^−1^ except for some EOs that are moderately toxic to very toxic such as boldo, cedar and pennyroyal with LD50 values of 130, 830 and 400 mg kg^−1^ [128]. Reports of allergenic potential have been made regarding the use of cinnamon and citronella oil [129,130]. Regarding economic considerations, areas of production are increasing every year and decreasing the prohibitive cost of EOs. With controversial products being removed from the market, such as the sprout-preventing chemical chlorpropham (CIPC), alternative products such as EOs are expected to rise. Techno-economic assessments are still lacking regarding a large number of applications. These evaluations combining efficacy, plant safety and social and environmental impacts should clarify many opportunities for the application of EOs [131].

To conclude, the use of EOs for sustainable agricultural practices seems promising, and extensive research will probably clarify or deny their relevance in diverse applications. Due to their inherent characteristics, the pest control properties are usually very transitory and less effective than synthetic products. However, EOs can be an efficient alternative to conventional plant protection products when properly formulated and integrated with other pest management strategies.

## Figures and Tables

**Figure 1 foods-09-01291-f001:**
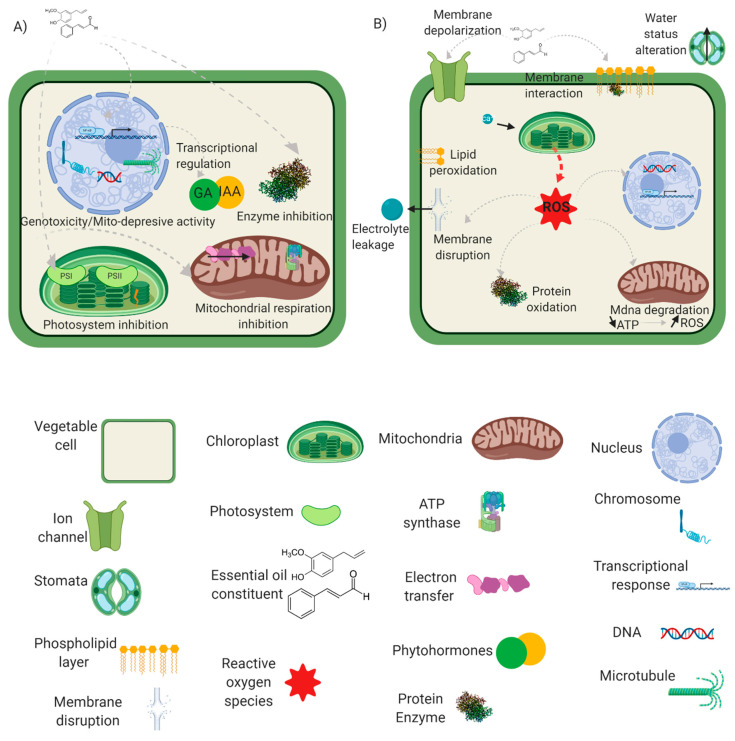
Mode of action of essential oil at the cellular level. (**A**) Photosynthesis and mitochondrial respiration inhibition, microtubule disruption and genotoxicity, enzymatic and phytohormone regulation. (**B**) Water status alteration, membrane properties and interactions, reactive oxygen species induction.

**Figure 2 foods-09-01291-f002:**
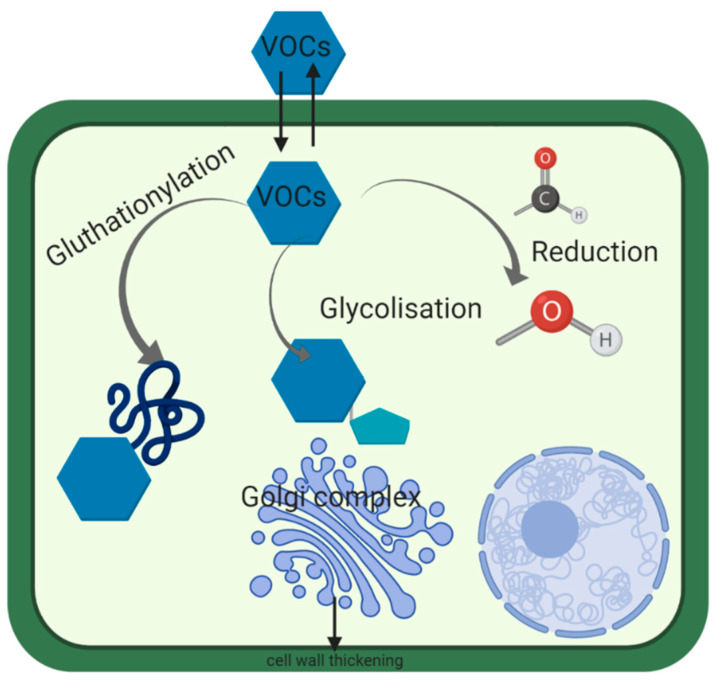
Sequestration and biotransformation of exogenous volatile organic compounds (VOCs) in plant.

**Table 1 foods-09-01291-t001:** Phytotoxic properties of essential oils or constituent solutions in diverse application mode/rate.

Mode of Action	Essential Oils or Constituents (Concentration)	Application Mode (Time)	Plant Target	Observation	Ref
Water status alteration	Camphor (10 mg/L) menthol (5 mg/L)	Vapor exposure (for 24 to 96 h)	*A. thaliana*	Scanning electron microscopy, transpiration, PCR, western blot	[25]
	Camphor (10 mg/L)	Vapor exposure (for 24 to 96 h)	*A. thaliana*	Real time PCR, in vivo cytoskeleton visualization	[26]
	Clove oil (2.5%) eugenol (1.5%)	Sprayed at 50 mL/m^2^	Broccoli, lambsquarte, pigweed	Membrane integrity (EL), spray solution retention	[27]
	Citral (1200–2400 μM)	Watered every 2 day (25 mL per pot)	*A. thaliana*	Water/osmotic potentials (Ψw/Ψs), pigment, protein, anthocyanin, stomata density	[28]
	Trans-caryophyllene (450–1800 µM)	Watering (25 mL/pot) or spraying (15 mL/pot)	*A. thaliana*	Chlorophyll a fluorescence, osmotic potential, MDA, pigment, proline, protein and element content	[29]
Membrane properties and interaction	*Mentha piperita* (5–900 ppm)	Perfusion	*Cucumis sativus*	Root segment membrane potential determination	[31]
	*C. zeylanicym C. winterianus* (3%)	Sprayed (10 L/m^2^)	*A. thaliana*	Herbicide tests + in silico approach	[35]
	1,8-cineole, thymol, menthol, geraniol, camphor (21.7, 2.0, 1.9, 2.5, 7.4 mg/L)	Vapor exposure	*Zea mays*	Lipid, peroxide and lipid peroxidation	[36]
				Sterols and phospholipid fatty acid (PLFA) composition	[37]
Reactive oxygen and nitrogen species induction	α-pinene (1.36–136 mg/mL)	Vapor exposure in petri dish for 3, 5 and 7 days	*C. occidentalis, A. viridis, T. aestivum, Pisum sativum, Cicer arietinum*	EL, MDA, H_2_O_2_, proline, ROS scavenging enzymes (SOD, APX, GPX, CAT, GR)	[50]
	β-Pinene (0.02–0.80 mg/mL)				[42]
	β-pinene (1.36–13.6 µg/mL)	Vapor exposure for 4 to 24 h	Wheat seed	H_2_O_2_, O^2−^, MDA, ROS scavenging enzymes, LOX	[43]
	Citronellol (50–250 μM)	Watered for 24, 48 and 72 h	Wheat seed	MDA, EL, CDs, LOX, In situ histochemical analyses	[44]
	*P. benghalensis* (0.25–2.5 mg/mL)	Vapor exposure	*Avena fatua Phalaris minor*	H_2_O_2_, O^2−^, MDA, CDs, EL, ROS scavenging enzymes	[45]
	*Monarda didyma* (0.06–1.25 µg/mL)	Vapor exposure for 5 days	Weed seed	H_2_O_2_, MDA	[46]
	*Artemisia scoparia* (0.14–0.70 mg/mL)	Vapor exposure for 5 days	Wheat seed	O^2−^, H_2_O_2_, proline, root oxidizability, cell death	[47]
	*Heterothalamus psiadioides* (1–5 µL)	Vapor exposure in petri dish for 7 days	*A. thaliana*	Histochemical detection of H_2_O_2_	[48]
Photosynthesis inhibition	β-pinene (135 µM)	Applied to organelles suspension	Chloroplast (*Spinacia oleracea*)	O_2_, protein, chlorophyll, electron microscopy	[51]
	β-pinene (945 µM)	Applied to organelles suspension	Chloroplast (*Cucurbita pepo*)	O_2_, protein, chlorophyll, Gel electrophoresis and immunoblotting	[52]
	β-pinene (0.02–0.80 mg/mL)	Vapor exposure for 3, 5 and 7 days	*Oryza sativa*	Chlorophyll, protein, carbohydrate, proteases, α- and β-amylases, POD, PER	[53]
	*Cymbopogon citratus* (1.25–10% (*v*/*v*))	Foliar sprayed at 1000 L ha^−1^	Barnyardgrass	Chlorophyll a, b and carotenoid, EL, MDA	[54]
Photosynthesis inhibition	*Hyptis suaveolens* (1–5% (*v*/*v*))	Foliar sprayed (10 mL/plant)	*Oryza sativaE. crus-galli*	Total chlorophyll content, cell viability, Cytogenetic analysis	[55]
	Farnesene (0–1200 μM)	Grown in medium for 14 days	*A. thaliana*	Root gravitropism, structural studies, electron microscopy, O^2−^, H_2_O_2_, microtubule, ethylene, auxin	[56]
	*Artemisia fragrans* (0.5, 1, 2 and 4%)	Spraying (100 mL/ pot) for 5 days	*Convolvulus arvensis*	Chlorophyll a fluorescence, chlorophyll, ROS scavenging enzymes, H_2_O_2_, MDA	[57]
	Clove oil (2.5%), eugenol (1.95%)	Covered by solutions	Broccoli	Chlorophyll a fluorescence imaging at 20, 40 and 60 min	[58]
	*Origanum vulgare* (0–500 μL/L)	Grown in medium for 10 days	*A. thaliana*	Chlorophyll a fluorescence, chlorophyll, protein, MDA, Ionomic, metabolomic	[59]
Mitochondrial respiration inhibition	1,8-cineole (6 mM)	Apply to organelle	*A. fatua*	O_2_ consumption	[60]
	Juglone (10 mM)	Bathed in dark for 30 min	Soybean cotyledons	O_2_ consumption and isotope fractionation	[61]
Mitochondrial respiration inhibition	α-pinene, camphor, eucalyptol and limonene (0.1–10 mM)	Vapor exposure/apply to organelle	Maize	Protein, seed germination, growth test and oxygen uptake	[62]
	α– pinene (50–500 µM)	Grown in medium for 10 days	Coleoptiles and primary roots of maize	O_2_ consumption, mitochondrial ATP production	[63]
	Pulegone, menthol, menthone (0–1500 ppm)	Foliar sprayed	Cucumber seeds (roots segments, mitochondria)	O_2_ uptake, mitochondrial respiration	[64]
	Camphor, 1,8-Cineole, Limonene, α–pinene (0–500 µM)	Apply to organelle suspension	Corn and soybean	Mitochondrial respiration	[66]
	1,8-cineole (0–2000 µM)	Vapor exposure	*N. tabacum* (seeds)	Growth, protoplasts proliferation, starch accumulation of BY-2	[68]
Microtubule disruption and genotoxicity	Citral (0–1.0 μL)	Vapor exposure	*A. thaliana*	Microscopy, in vitro polymerization of microtubules	[70]
	Citral (0–1.200 μM)	Grown inmedium for 14 days	*A. thaliana*	Ultra-structural, pectin and callose staining, mitotic indices, ethylene, auxin	[71]
	Limonene, citral, carvacrol, pulegone (4.6–9.2 μmol/20 mL)	Vapor exposure for 0, 15, 30 and 60 min	*A. thaliana*	Membrane, microtubules, F-actin, (confocal microscopy), *in Planta* monoterpene concentrations	[73]
	Menthone	Vapor exposure	*Tobacco BY-2A. thaliana*	GFP-tagged markers for microtubules and actin filaments	[74]
	*Schinus molle Schinus terebinthifolius*	Vapor exposure 0.1 mL for 72 h	*Allium cepa, Lactuca sativa*	Cytogenetic assay	[75]
	*Citrus aurantiifolia* (0.10–1.50 mg/mL)	Vapor exposure (10 mL) for 3–24 h	*Avena fatua*, *E. crus-galli, Phalaris minor*	Phytotoxicity: dose-response assay, cytotoxicity (*Allium cepa*)	[76]
	*Plectrantus amboinicus* (0–0.120% *w*/*v*)	Vapor exposure for 48 h	*Lactuca sativa Sorghum bicolor*	Germination speed index, percentage of germination	[77]
	*Mentha longifolia* (10–250 μg/mL) (0.5–5%)	Vapor exposure Foliar sprayed (5 mL/pot)	*Cyperus rotundus, E.crus-galli, Oryza sativa*	Germination, root length, coleoptile length, chlorophyll, cytotoxicity assay (*Allium cepa*)	[78]
Microtubule disruption and genotoxicity	*Nepeta nuda* (0.1–0.8 µL/mL)	Vapor exposure (10 mL) for 7 days	*Zea mays*	Randomly amplified polymorphic DNA, quantitative analysis of proteins	[79]
	*Salvia leucophylla* (0–1300 µM)	Vapor exposure for 4 days	*Brassica campestris*	DAPI-fluorescence microscopy, immunofluorescence microscopy, DNA Synthesis Activities	[80]
	*Vitex negundo* (0.1–2.5 mg/mL)	Vapor exposure (12 mL)	*Avena Fatua, E. crus-galli,* Onion bulbs	Phytotoxicity, cytoxicity	[81]
	*S*-carvone (125 µL)	Vapor exposure (several days)	*Solanum tuberosum*	Potato sprout growth, HMGR activity, membrane protein composition, transcription activity	[82]
Phytohormones	*R*/*S*-carvone (25–125 µL)	Vapor exposure (several days)	*Solanum tuberosum*	Growth inhibition, carvone and conversion products in potato sprouts	[83]
	Peppermint oil (0.1% (*v*/*v*))	Vapor exposure	*Solanum tuberosum*	Potato sprout growth, protein extraction, enzyme activity, semi quantitative RT-PCR for potato α–amylase	[84]
	Ten monoterpenes (0.5–2 mM)	Vapor exposure (6 mL) for 9 days	*Silybum marianum*	carbonic anhydrase activity	[87]
	Farnesene (250 μM)	Grown in medium for 14 days	*A. thaliana*	Root anatomy/meristem size, mitotic indices, quantitative PCR, auxin gradient and polar transport	[88]

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
