# Peer review of "Phytotoxicity of Essential Oils: Opportunities and Constraints for the Development of Biopesticides. A Review"

_foods, 2020, doi:10.3390/foods9091291_

Round 1

Reviewer 1 Report

The present manuscript deals with a literature review regarding an interesting topic to the field. It is well structured and the number of references is high enough and they are well discussed. 

I only have some minor aspects to be improved before its publication: 

In figure 1, authors should edit the second part of the figure (that about the explanation of the designs) to be clearer. Perhaps by putting closer the name with the drawing, to avoid misunderstandings.

Page 5, lines 158-169. Authors say that numerous publications describe that fact, but they only reference one article (that could be a review). Please, rewrite the sentence or include more references that corroborate the fact. This also happens in page 7, lines 274-275.

In my opinion, section 3 is very short. I suggest to include some more explanations regarding this topic.

I would also include a table in which a general overview of the references employed and their impacts could be easily seen. 

Author Response

Thank you for your precious advice,

Please see the attachment for corrections

Reviewer 2 Report

Plant essential oils (EO) become more important in agriculture because of the risk in the extensive use of pesticides. In the present review authors discuss the opportunities as well as the constrains of using EOs as biopesticides. The focus of the review is on the molecular mechanisms underlying the phytotoxicity of EOs, and that is novel-

The paper is a topical and important contribution to our knowledge on practical use of plant EOs. The contribution is well structured and provided with memorable figures. There are only few typographical errors in the text:

  • line 138: Mode of ...
  • line 180: has been observed
  • line 343 and others: use acronym for species name after first mentioned
  • line 521: better say "in Table 1"

Unfortunately, References are written very poor and have to be rewritten completely: species names in italics, paper titles in lowercase letters, proper acronyms for journal titles, many references are incomplete etc.

Author Response

(The authors gave the same response as above.)

Reviewer 3 Report

The review entitled “Phytotoxicity of essential oils: opportunities and constraints for the development of biopesticides. A review ”deals with a topic that is truly topical and of considerable interest both for researchers and for those who work in the research and development field of chemical product companies and who want to develop products in the bio field. However, despite the title being very promising, I feel the work lacks content. I would like to quote a recently published review on Foods (Foods 2020, 9, 365; doi: 10.3390 / foods9030365) which is an excellent example of how a review of this type should be set up.

For this reason I believe that the manuscript needs a thorough revision. Allow me to suggest some notes to follow:

Graphical abstract: the right part is not clear, as the writings are not read. Given that the figure concerning the cellular and target mechanisms also appears in the text (Figure 1), one might think of eliminating it from the graphical abstract.

Review all references (the References caption is also missing): many do not respect the style of the magazine (eg note 5, 49…. The name of the journal is missing… note 60 is missing the name of the journal, pages, etc and moreovere….).

Figure 1: Essential oil constituant can be replaced with constituent

The format of the captions must be changed according to the guidelines of the magazine:

Figure 1. This is a figure, Schemes follow the same formatting. If there are multiple panels, they should be listed as: (a) Description of what is contained in the first panel; (b) Description of what is contained in the second panel. Figures should be placed in the main text near to the first time they are cited. A caption on a single line should be centered.

Line 214: citronelal must be replaced with citronellal, and citronelol with citronellol

Line 218, 336: the same above

Line 226: 1.8-cineole must be replaced with 1,8-cineole

Line 227-228: What changes were highlighted by this study? In my opinion, in a review, the results obtained from the aforementioned studies should be explored a little more. It applies to this paragraph as to all the others in the manuscript.

Paragraph 2.4: we move from note 38 to 42, 43 and then return to 41. Note 39 is found in the next paragraph, note 40 does not appear in the text ... I advise authors to carefully review the entire manuscript.

Line 284-285: A thaliana must be replaced with A. thaliana

Line 342: tobaccoBY-2 must be replaced with tobacco BY-2

Line 384: asparginine must be replaced with asparagine

The manuscript is a review on essential oils but it is all written in a very general way, there are few examples of specific essential oils. The actions of some components of essential oils are reported, but without specifically talking about concentrations Since the title mentions opportunities for the development of biopesticides based on essential oils, I think it is also essential to talk about the concentration-effect relationship for each mechanism in relation to the type of essential oil used, and not focus the entire review on the individual components (among other things, very few examples reported). Only in the conclusions there is a little mention of concentration limits (LD50) but it would be appropriate that in a review of this type the various mechanisms of action were associated with the concentration of possible use.

Author Response

(The authors gave the same response as above.)

Reviewer 4 Report

This work is of interest.

1) Please, correct any typos.

2) Paragraph 2.4 Reactive oxygen and nitrogen species induction: this paragraph must be increased with further examples.

3) Paragraph 2.5 Photosynthesis inhibition: this paragraph must be increased with further examples.

4) Paragraph 2.6 Mitochondrial respiration inhibition: this paragraph must be increased with further examples.

5) Table 1 must be improved.

Author Response

(The authors gave the same response as above.)

Round 2

Reviewer 3 Report

I thank the authors for having responded in detail to each of the requests / suggestions proposed. I believe that in this version the manuscript has been enriched with parts that in my opinion were fundamental for publication.